# KDR (VEGFR2) Genetic Variants and Serum Levels in Patients with Rheumatoid Arthritis

**DOI:** 10.3390/biom9080355

**Published:** 2019-08-09

**Authors:** Agnieszka Paradowska-Gorycka, Barbara Stypinska, Andrzej Pawlik, Damian Malinowski, Katarzyna Romanowska-Prochnicka, Malgorzata Manczak, Marzena Olesinska

**Affiliations:** 1Department of Molecular Biology, National Institute of Geriatrics, Rheumatology and Rehabilitation, Spartańska 1, 02-637 Warsaw, Poland; 2Department of Physiology, Pomeranian Medical University, 70-111 Szczecin, Poland; 3Department of Pharmacology, Pomeranian Medical University, 70-111 Szczecin, Poland; 4Department of Connective Tissue Diseases, National Institute of Geriatrics, Rheumatology and Rehabilitation, 02-637 Warsaw, Poland; 5Department of Pathophysiology, Warsaw Medical University, 02-637 Warsaw, Poland; 6Department of Gerontology and Health Promotion, National Institute of Geriatrics, Rheumatology and Rehabilitation, 02-637 Warsaw, Poland

**Keywords:** angiogenesis, inflammation, gene polymorphisms, KDR, rheumatoid arthritis, protein level

## Abstract

We investigated kinase insert domain-containing receptor (KDR) polymorphisms and protein levels in relation to susceptibility to and severity of Rheumatoid Arthritis (RA). 641 RA patients and 340 controls (HC) were examined for the rs1870377 KDR variant by the polymerase chain reaction (PCR)-restriction fragment length polymorphism (RFLP) method and for rs2305948 and rs2071559 KDR single nucleotide polymorphisms (SNPs) by TaqMan SNP genotyping assay. KDR serum levels were determined by enzyme-linked immunosorbent assay (ELISA). The rs1870377 KDR variant has shown association with RA under the codominant (*p* = 0.02, OR = 1.76, 95% CI = 1.09–2.85) and recessive models (*p* = 0.019, OR = 1.53, 95% CI = 1.07–2.20). KDR rs2305948 was associated with RA under the dominant model (*p* = 0.005, OR = 1.38, 95% CI = 1.10–1.73). Under the codominant model, the frequency of the rs2071559 TC and GG genotypes were lower in RA patients than in controls (*p* < 0.001, OR = 0.51, 95% CI = 0.37–0.69, and *p* = 0.002, OR = 0.57, 95% CI = 0.39–0.81). KDR rs2071559 T and rs2305948 A alleles were associated with RA (*p* = 0.001, OR = 0.60, 95% CI = 0.45–0.81 and *p* = 0.008, OR = 1.71, CI = 1.15–2.54). KDR rs2305948SNP was associated with Disease Activity Score (DAS)-28 score (*p* < 0.001), Visual Analog Scale (VAS) score (*p* < 0.001), number of swollen joints (*p* < 0.001), mean value of CRP (*p* < 0.001). A higher KDR serum level was found in RA patients than in HC (8018 pg/mL versus 7381 pg/mL, *p* = 0.002). Present results shed light on the role of KDR genetic variants in the severity of RA.

## 1. Introduction

Rheumatoid arthritis (RA) is not only autoimmune disease, but it is also a member of the “angiogenic family of diseases” due to active tissue neovascularization [1,2]. Although the molecular mechanisms promoting neovascularization/angiogenesis in patients with RA have not been clearly identified, it is accepted that this process may lead to immune activation as well as the progression of synovitis [3,4,5]. Synovial angiogenesis is clearly a feature of both early and late stages of RA, with vascular endothelial growth factor (VEGF) playing a pivotal role not only in angiogenesis but also in inflammation [6,7,8]. VEGF acts through interaction with the one or both high-affinity tyrosine kinase receptors, VEGF receptor-1(VEGFR-1) and VEGF receptor-2 (VEGFR-2) [9,10,11]. VEGFR2, also known as fetal liver kinase-1 (Flk-1) in mice and kinase insert domain-containing receptor (KDR) in humans, is a major mediator of angiogenic, mitogenic, and vascular permeability activity [12,13,14]. In addition, KDR expression increased in response to hypoxia, which is a key inducer of *VEGF* gene transcription [5]. An increased VEGF/KDR signaling not only enhanced angiogenesis but also play a key role in atherosclerosis as well as a chronic inflammation [15,16].

In response to pro-inflammatory cytokines, the VEGF/KDR system expression, in serum as well as the synovial fluid of patients with RA, is upregulated and has shown association with disease severity and activity [11,17,18,19,20,21,22]. Thus, mechanisms involved in VEGF/KDR system control may be critical in autoimmunity and in RA development and/or course. In our earlier study, we have shown that *VEGF* single nucleotide polymorphisms (SNPs), as well as serum VEGF levels, may be associated with the onset of RA [23]. In order to better define the role of genetic factors essential for the development of RA and its clinical manifestation, it is necessary to recognize an individual’s likely developing rheumatoid arthritis. Studies on SNPs that affect the risk of developing rheumatoid arthritis are numerous. However, there are few reports on how gene polymorphisms may modulate the severity of RA. Some previous studies have shown that LRP5 A1330V, LRP5 V667 M, RAD51B rs911263, TLR10 I473T, IL-6 rs4453032 SNPs were associated with erosion in RA patients [24,25,26,27]; LRP5 V667 M, NLRP3 rs10754558 and CARD8 rs2043211 polymorphisms were associated with Health Assessment Questionnaire (HAQ) scores [24,28]; and FLT-1 rs7324510 and AFF3 rs11676922 SNPs were associated with VAS score, DAS-28 score, and ExRA presence [29,30].

In this study, we carry out a candidate gene study of the correlation between KDR and the susceptibility to and severity of RA. We selected three functional SNPs located in the *KDR* gene and analyzed their association with RA in the Polish population. In a subgroup of 272 patients with RA, we also determined the KDR serum levels, which were next correlated with the severity of RA, RA phenotype, and *KDR* genotypes. Moreover, we also examined whether the *KDR* gene SNPs and serum KDR levels are related to the development of cardiovascular disease (CVD) in our RA patients.

## 2. Results

### 2.1. Characteristics of the Study Population

Our study population included 641 patients with RA and 340 healthy subjects as a control group (217 females and 123 males aged between 18 and 63 years). Demographic and clinical characteristics of our RA patients, collected at the time blood sampling, is presented in Table 1. Among all RA patients, 69% were RF positive, and 80% had the Anti-Citrullinated Peptide Antibody (ACPA). Table 2 was shown characteristics of RA patients with and without cardiovascular events (CVD). The symptoms of CAD was found in 13% of patients, myocarditis in 3% of patients, and hypertension in 36% of patients; all this evidence was classified as CVD. In the present study, we observed that RA patients with CVD were older (62 versus 53 years; *p* < 0.001) and had a higher activity of disease than RA patients without CVD. However, after Bonferroni correction for multiple testing, the parameters such as the mean value of ESR and CRP, HAQ score, and creatinine levels did not show significant differences between RA patients with CVD and without CVD.

### 2.2. Information about KDR Gene SNPs as Well as Distribution of the KDR Genetic Variants in RA Patients and Controls

Two nonsynonymous coding SNPs at position rs2305948 (V297I) and rs1870377 (H472Q) are located in the extracellular fourth and fifth immunoglobulin-like domains, which are important for binding the VEGF to KDR [31,32]. KDR rs2071559 (−604T/C) SNP is located in the 5′ untranslated region (5’-UTR) of the gene, and it may affect the KDR transcription. This genetic variant may probably change the structure of the binding site for the transcriptional factor E2F (involving in cell cycle regulation) in the *KDR* gene promoter region [31,33].

The minor allele frequency (MAF) of the two chosen SNPs, rs2305948 and rs2071559 in our cohorts were similar to those in the western European ancestry and Utah residents of northern (HapMap database; Appendix A). Furthermore, the *KDR* minor rs1870377 T allele frequency was higher in Polish RA patients (32%) as well as controls (27%) than in other European populations (21%). Using the selected sample size of 641 RA patients and 340 healthy controls at the significance level of 0.05, the power of our study was 96% for rs1870377, 87% for rs2305948 and 98% for rs2071559.

Next, we checked an association between *KDR* genetic variants and susceptibility to RA. The distributions of the *KDR* rs1870377, rs2071559 and rs2305948genotypes among RA patients and healthy subjects, were presented in Table 3. Three genetic models such as codominant, dominant and recessive we used to assess the possible association between *KDR* SNPs and RA risk.

We observed a significant difference in genotype frequency of *KDR* rs1870377 and rs2071559 polymorphisms between RA patients and control groups. The frequency of the rs1870377TT genotype was higher in RA patients compared with controls under the codominant and recessive model (*p* = 0.02 and *p* = 0.019, respectively). Frequency of the rs2071559TC and rs2071559CC genotypes were significantly lower in RA patients compared to the healthy subjects (*p* < 0.001 and *p* = 0.002, respectively). Moreover, under the dominant model, the frequency of the TC + CC genotype was also lower in RA patients than in controls (*p* < 0.001). In the case of the rs2305948 *KDR* variant, we found that under the dominant model the frequency of the GA + AA genotype was significantly higher in RA patients than in healthy subjects (*p* = 0.005).

Our analysis revealed that KDR gene rs2305948 A allele was associated with significantly increased risk of RA (*p* = 0.008) than the rs2305948 G allele in the Polish population. Furthermore, KDR gene rs2071559 C allele was more frequently observed in healthy subjects than in RA patients (*p* = 0.001).

### 2.3. Genetic Effects of the KDR Haplotypes on the RA risk

The linkage disequilibrium (LD) map of the *KDR* rs1870377, rs2305948, and rs2071559 polymorphisms was generated using the SHEsis program. The analysis results revealed weak LD between examined KDR gene polymorphism, as the r2 value between all of them is considerably low Figure 1.

Next, we investigated whether haplotypes created by the examined polymorphisms located in the *KDR* gene may have an impact on the RA risk in our population. Six major *KDR* haplotypes with frequency >0.03%, were found in patients with RA as well as in controls (Table 4). The most common haplotypes identified in our Polish population were rs1870377A/rs2305948G/rs2071559C, which were estimated with frequencies of 37% in RA patients and 49.8% in healthy subjects. This association is translated into a protective effect (OR = 0.601, CI = 0.494–0.731; *p* < 0.001). Another often observed haplotype in our population was rs1870377A/rs2305948G/rs2071559T. This AGT haplotype was observed in 24% of RA patients and 17% of healthy subjects, and it may be associated with risk of RA (OR = 1.506, CI = 1.176–1.927; *p* = 0.001). For the other haplotypes, we found no differences in haplotype distribution between patients with RA and controls.

### 2.4. Association between KDR rs2305948 Polymorphism and Clinical Phenotype of Rheumatoid Arthritis

Because *KDR* rs1870377, rs2305948, and rs2071559-genetic variants showed association in the pooled analysis of Polish subjects, and because of our hypothesis that *KDR* may be a good candidate gene to play a part in RA inflammatory processes, we decided to carry on analysis whether *KDR* SNPs may have an impact on RA phenotype. Our data showed that only the *KDR* rs2305948 polymorphism was significantly associated with the RA phenotype. As shown in Table 5, a detailed genotype-phenotype comparison was conducted among RA patients in relation to demographic, clinical, as well as biochemical parameters. After Bonferroni correction, our analyses indicate significant correlation of the *KDR* rs2305948GG variant with the number of swollen joints (*p* < 0.001), CRP (*p* < 0.001), VAS score (*p* < 0.001), and DAS-28 score (*p* < 0.001). Our analysis did not show significant relationship between KDR rs2305948+889 GG variant and HAQ score (*p* = 0.027), Larsen score (*p* = 0.03), ESR (*p* = 0.006), as well as creatinine level (*p* = 0.027).

The carriers of this *KDR* rs2305948GG genotype had mostly higher laboratory parameters compared to RA patients with the KDR rs2305948A allele. In contrast, we also observed that the number of RA women with *KDR* rs2305948A allele was higher than the number of RA women with *KDR* rs2305948GG variant (*p* < 0.001). The influence of *KDR* rs1870377 and rs2071559 polymorphisms on clinical symptoms of RA showed a significant correlation with RF presence and with the number of women, respectively. RF was more frequently observed in RA patients with the *KDR* rs1870377T allele than in RA patients with the *KDR* rs1870377AA genotype–this association was not significant after Bonferroni correction (*p* = 0.03, Appendix A). Furthermore, the number of RA women with the *KDR* rs2071559C allele was higher than the number of RA women with the *KDR* rs2071559TT genotype—this association was not significant after Bonferroni correction (*p* < 0.05, Appendix A).

### 2.5. KDR Protein Level in Healthy Subjects and Patients with RA. Association with RA Clinical Phenotype

Because KDR plays a significant role in angiogenesis and inflammation, we decided to correlate the KDR serum levels not only with susceptibility to RA but also with the severity of RA. The KDR serum level was assessed in 272 RA patients and 290 healthy individuals. We observed that KDR serum level was higher in RA patients comparing with healthy subjects (8018 pg/mL versus 7381 pg/mL, *p* = 0.002; Figure 2).

In the next step, we divided our RA patients into two groups: in group I, was RA patients with high disease activity (DAS-28 ≥ 5.0), RF-positive, ACPA-positive, and CVD presence; while group II contained the RA patients with low disease activity and without ACPA, RF, and CVD. The KDR serum level was compared between both these groups (Table 6). We found no association between KDR serum level and RA phenotype. Although, we observed some tendency to differences in KDR serum level between both groups. We observed that the KDR serum level was higher in RA men comparing with RA women (*p* = 0.093). Moreover, RA patients with a number of swollen joints <3 as well as a number of tender joints <7 had a higher KDR serum level than RA patients from group I (both *p* = 0.081).

### 2.6. KDR Genetic Variants with Respect to KDR Protein Levels

The last step of our analysis was to determine the impact of *KDR* gene SNPs on KDR protein levels in RA patients and healthy subjects. First, we conducted a comparative analysis between RA patients and controls in relation to *KDR* rs1870377, rs2071559, and rs2305948 polymorphisms (Table 7). KDR protein levels in RA patients with rs1870377AA, rs2071559CT, and rs2305948GA genotypes were significantly higher compared to controls with the same *KDR* genotypes (*p* = 0.001, *p* = 0.002, and *p* = 0.001, respectively).

Next, we determined the relationship between KDR protein levels in RA patients according to *KDR* gene polymorphisms (Figure 3). We found some differences among RA patients in relation to *KDR* gene SNPs. In RA patients with the *KDR* rs1870377AA genotype, the KDR serum levels were significantly higher compared with RA patients with *KDR* rs1870377AT or rs1870377TT genotypes (*p* = 0.03). RA patients with rs2071559CC genotype had the lowest KDR serum levels comparing with RA patients with rs2071559CT or rs2071559TT genotypes (*p* = 0.009). Furthermore, RA patients with rs2305948GA genotype had the highest, and RA patients with rs2305948GG genotype had the lowest KDR serum levels (*p* = 0.02).

## 3. Discussion

We investigated, for the first time to our knowledge, the potential involvement of the *KDR (VEGFR2)* gene polymorphisms in rheumatoid arthritis susceptibility and severity. In the present study, we observed that the *KDR* rs1870377TT genotype, rs2305948A allele and rs1870377T/rs2305948A/rs2071559T haplotypes showed significant association with risk of RA. Furthermore, the *KDR* rs2071559C genetic variant and rs1870377A/+889 G/rs2071559C haplotype may play a protective role in the development of RA in our population. Additionally, in our study, we also found that the *KDR* rs2305948GG genotype showed a significant positive association with disease severity.

One of the key measures of severity of rheumatoid arthritis is angiogenesis playing a central role in the inflammation and joint damage [4,5,34,35]. The best-characterized system regulating the angiogenesis in the rheumatoid joint is VEGF-VEGFRs [12]. In RA patients, KDR expression is upregulated in synovial tissue [3], but during hypoxic conditions, plasma membrane levels of KDR are depleted [36]. Some previous reports have described the *KDR* mRNA expression levels in RA patients [11,35], but this study is the first study that examined the KDR protein levels in the blood of RA patients. We have shown that KDR protein levels in patients with RA were higher than in healthy subjects, reflecting not only the inflammation but also angiogenesis in RA patients. Binding of VEGF to KDR initiates the activation of the phospholipase C-gamma (PLC-*γ*)/mitogen activated protein kinase (MAPK) pathway and promotion of endothelial cell proliferation [37,38] as well as PLC-*γ*/phosphatidylinositol 3′ kinase (PI3K)/AKT pathway leading to cell migration and vascular permeability [37]. On the other hand, we also observed that the KDR serum levels were higher in RA patients with lower disease activity (group II, Table 6). We can speculate that this situation may be caused by increased hypoxia in the inflamed joints of the RA patients from group I. Increased hypoxia through selective activation of anti-angiogenic molecule expression, such as VEGFR1 (FLT-1), acts as a negative regulator for VEGF activity by KDR.

Limited data are available on the effect of polymorphisms of angiogenic factors such as *KDR* on arthritic pathophysiology, despite angiogenesis playing an important role in the RA pathogenesis. In contrast, some studies have shown that polymorphisms located in the *KDR* gene are associated with the risk of several human diseases such as chronic myeloid leukemia, breast cancer, glioblastoma, colorectal cancer, cardiovascular disease, or type 2 diabetes [31,39,40,41,42,43,44,45,46]. In rheumatoid arthritis, research focuses more on determining the angiogenic factors’ mRNA expression than on the analysis of genetic variants. Determining the genetic predisposition for RA is difficult due to the number of disruptive factors. That is why analysis of genetic variants of factors influencing the susceptibility to or severity of RA is so important; to determine new factors that predispose to RA. In this study, we defined three *KDR* genetic variations in Polish RA patients and assessed the phenotypic impact of these functional variants. We found that two of these three examined SNPs have a significant effect on RA susceptibility. The *KDR* rs1870377T variant, located in exon 11, decreased the KDR serum expression levels and, at the same time, increased the risk of RA in our population. In contrast, the *KDR* rs2071559C variant, located in the 3’-UTR region, also decreased the KDR serum expression levels and may play a protective role in the development of RA in our population. Furthermore, we demonstrated that one of these three studied genetic variants had a strong effect on RA phenotype. The *KDR* rs2305948GG genotype, located in exon 7, decreased KDR serum expression levels and lead to higher disease activity in our patients. We suggest that the decreased KDR protein expression associated with a regulatory SNP at position rs2071559C appears to be due to a gene transcription mechanism. It is possible that this polymorphism may decrease identification, as well as binding, of VEGF to KDR, which can affect the gene expression and inhibit angiogenesis leading to lower risk of RA in our population. Moreover, in our opinion, the decreased KDR protein expression in RA patients, associated with two functional *KDR* gene SNPs at position rs1870377T and rs2305948GG, appears to be due to lower affinity of VEGF to KDR, which results in dysfunctional KDR, higher affinity VEGF for FLT-1 or hypoxia-inducible factor (HIF)-1α, intensified angiogenesis, and a higher risk of RA, as well as higher disease activity in our patients. Also, several genetic variants located in the VEGF-A gene have shown associations with rheumatoid arthritis, cancer, coronary artery disease, and chronic obstructive pulmonary disease. The rs1570360, rs699947, rs2010963, rs833070, and rs3025030 VEGF gene polymorphisms were associated with RA [23,47], cancer [48,49,50,51,52], heart disease [53], chronic obstructive pulmonary disease [54,55], or type 2 diabetes [56]. While the VEGF-C gene polymorphisms were not examined in patients with RA, they were examined in patients with cancer and Kawasaki disease. VEGF-C rs7664413, rs2046463, and rs1485766 SNPs have shown association with cancer [57,58,59], and -634 G/C have shown association with Kawasaki disease [60].

In our study, we observed that not all examined polymorphisms were concordant with HWE; this can be treated as a limitation. Deviations from HWE can be very informative; they could imply a sampling bias, inbreeding, genetic drifting, mistyping of genotypes, differential survival of marker carriers, ethnic differences, sample size, populations stratification or migration, and/or a combination of these reasons. Firstly, we checked for genotype call errors. The genotyping error minimization was achieved by randomLy repeating genotyping on 20% of selected samples (10% for RA patients, and 10% for healthy subjects), giving complete conformity of the result. Secondly, our sample size is relatively small; this may lead to genetic drift, which can result in a loss of polymorphism and drive the frequency of one allele to 1.

However, in the case of subjects, withdrawal from HWE, assuming that sources of errors have been eliminated, may indicate a genetic association and a connection of the locus with the disease.

In summary, our results suggested that examined *KDR* genetic variants may be associated with rheumatoid arthritis either by increased or decreased KDR protein expression levels. In addition, our study has shown that not only hypoxic condition but also gene polymorphisms, may change KDR expression levels and lead to different RA activity. Our findings may help in understanding the molecular genetics of angiogenesis as well as inflammation, not only in RA but also in other autoimmune diseases. Our results should be confirmed on larger sample sizes and in different populations. However, this is the first study supplying the evidence on the associations between *KDR* SNPs, KDR serum levels, and susceptibility to and severity of RA.

## 4. Materials and Methods

### 4.1. Study Population

RA patients included for this study were recruited from the National Institute of Geriatrics, Rheumatology, and Rehabilitation in Warsaw, Poland and from the Pomeranian Medical University in Szczecin, Poland. All patients with RA recruited for this study satisfied the criteria for RA. The inclusion criteria for RA patients were: all (male and female) RA patients who fulfill the 1987 American College of Rheumatology (ACR) or the 2010 EULAR/ACR criteria for RA, aged ≥ 18 years, and of Polish ethnicity. Exclusion criteria were a history of or current other inflammatory rheumatological or autoimmune disorders; malignancy; significant unstable or uncontrolled acute or chronic disease, which could confound the results of the study and/or current active infection. RA patients who qualified for the study had physical examinations and laboratory tests conducted. Demographic and clinical parameters such as age, gender, disease duration, number of tender and swollen joints, C-reactive protein (CRP), erythrocyte sedimentation ratio (ESR), platelets (PLT), creatinine, presence of rheumatoid factor (≥34 IU/mL), presence of anti-CCP antibodies (≥17 U/mL), disease activity score in 28 joints (DAS-28), visual analogue scale (range 0–100), Health Assessment Questionnaires (range 0–3), Larsen score, and extraarticular (ExA) manifestation were collected at the time of the clinical materials sampling.

The control groups were selected from healthy blood bank donor volunteers without a history of autoimmune/inflammatory diseases and cancers. Patients and healthy donors were from the same geographical area, and they had the same ethnicity as well as socioeconomic status.

Informed consent was obtained from all RA patients and healthy individuals included in the study. The study was approved by the Research Ethics Committee of the National Institute of Geriatrics, Rheumatology, and Rehabilitation (of 29 May 2014), and by the Research Ethics Committee of the Pomeranian Medical University. All procedures performed in this study were in accordance with the ethical standards of our Institute and with the 1964 Helsinki declaration and its later amendments or comparable ethical standards.

### 4.2. SNP Selection, DNA Extraction and Genetic Analysis

The KDR genetic markers were selected based on the previous publication and four assumptions: the SNPs with a MAF < 0.05 (<5%) were excluded, the position of SNPs in the KDR gene, possible functional effects, as well as association with other autoimmune diseases. Based on these assumptions for genotyping in our study, we selected three KDR genetic variants: rs2305948, rs2071559, and rs1870377. Genomic DNA was extracted from the blood of RA patients and healthy subjects using the QIAamp DNA Blood Mini Kit (Qiagen, Hilden, Germany) in accordance with the manufacturer’s instructions. The TaqMan allelic discrimination assay used to genotype *KDR* genetic variants were: C__22271999_20 (rs2305948) and C__15869271_10 (rs2071559). The reaction was performed using a Quant Studio 5 detection system (Life Technologies), and reaction conditions were as follows: denaturation at 95 °C for 10 min, followed by 40 cycles of denaturation at 92 °C for 15 s, and annealing and extension at 60 °C for 1 min. SNP genotyping was carried out using the TaqMan^®^ genotyping platform (Life Technologies, Carlsbad, CA, USA).

*KDR* rs1870377 variant was detected by the polymerase chain reaction (PCR)-restriction fragment length polymorphism (RFLP) method. Reaction mixture contained: 200 ng of genomic DNA, 10 pmol of each primer: forward 5′-TGC TTC CCT CCT GTA TCC TG-3′, reverse 5′-C CAT CCT TCC ATT AAA GAG AGA-3′, 0.25 mM of each dNTP and 1U HotStar Taq Polymerase (both Qiagen, Hilden, Germany), 1 × PCR buffer (containing 1.5µM magnesium chloride, Sigma, MO, USA). Reaction condition was as follows: 95 °C for 15 min, 35 cycles at 94 °C for 1 min, 49.7 °C for 1 min and 72 °C for 1 min, and a final extension at 72 °C for 10 min. The PCR product (392 bp) was digested with 1 µL of restriction enzyme AluI (Thermo Scientific (Fermentas, Finnzymes, Pierce, Abgene) Waltham, Massachusetts, USA). The results of the digestion were as follows: 392 bp product for allele T and 173 bp and 219 bp fragments for allele A.

### 4.3. Assay for Serum Levels of KDR

The serum concentration of KDR was determined using a standard sandwich enzyme-linked immunosorbent assay (ELISA) (R&D System, Abingdon, Oxon, UK) in accordance with the manufacturer’s instructions. For each analysis, we used 100 µL of serum. All samples were examined in duplicate. The mean value was used for statistical analysis. The minimum detectable dose (MDD) of KDR ranged from 1.0–11.4 pg/mL.

### 4.4. Statistical Analysis

The continuous variables distribution was checked by Shapiro–Wilk test. All continuous variables were not normally distributed; therefore, they are summarized as the median and interquartile range (IQR). The χ2 test or χ2 test with Yates’ correction (categorical variables) and U Mann-Whitney test (continuous variables) were used to compare the clinical/ serological parameters between groups. The association between SNPs and the risk of RA was estimated by unconditional logistic regression analysis under three genetic models, including codominant, dominant, and recessive models. Odds ratios (ORs) adjusted for sex and age and 95% confidence intervals (CIs) were calculated. A *p*-value of less than 0.05 was considered significant. All KDR genotype distribution was compared for Hardy–Weinberg Equilibrium (HWE) using the software HardyWeinberg Simulator (available at Institute of Human Genetics, Helmholtz Zentrum München, Germany). The power of the study was calculated using the Genetic Association Study (GAS) Power Calculator (http://csg.sph.umich.edu/abecasis/gas_power_calculator/index.htmL). The power was calculated for all examined KDR SNPs, taking into account sample size (641 cases and 340 controls), significance levels (0.05), RA prevalence (0.01), RA allele frequency (0.68 for rs1870377, 0.81 for rs2305948, and 0.52 for rs2071559), and genotype relative risk (1.5).

The presence of linkage disequilibrium (LD), as well as a coefficient (D′ and r2) for haplotypes, were carried out using the SHEsis software, available at http://analysis.bio-x.cn [61,62]. We used Bonferroni correction to adjust *p*-values for multiple measures; this correction was used (1) to determine clinical differences between RA patients with and without cardiovascular diseases, and (2) when we correlated KDR gene polymorphisms with the clinical phenotype of RA. Bonferroni-corrected α-level of *p* < 0.003 was considered statistically significant.

## Figures and Tables

**Figure 1 biomolecules-09-00355-f001:**
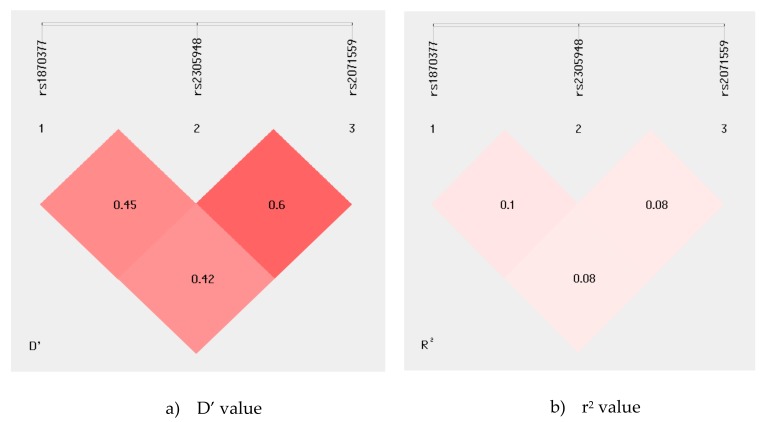
(**a**), D’ value; (**b**), r^2^ value. Linkage disequilibrium (LD) map of the KDR gene polymorphisms. The map illustrates the pairwise LD between KDR SNPs based on D′ and r^2^ values. Values approaching zero indicate the absence of LD, and those approaching 100 indicate complete LD.

**Figure 2 biomolecules-09-00355-f002:**
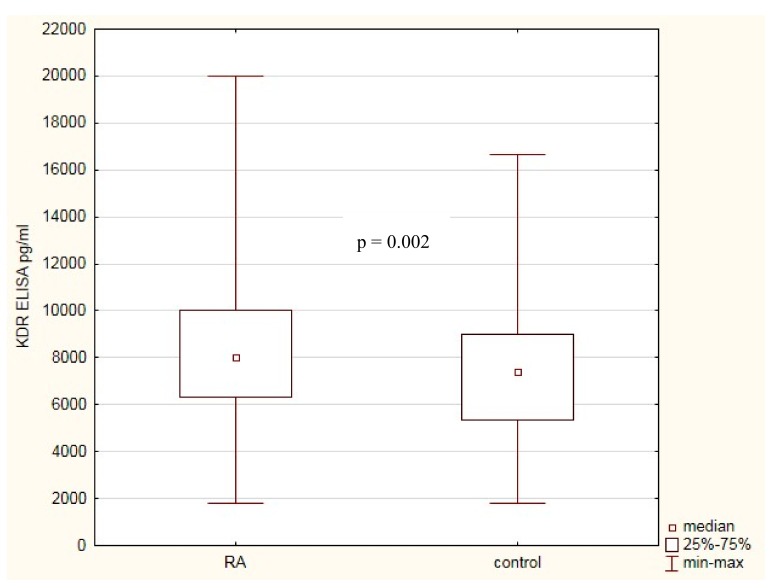
KDR serum concentrations in RA patients and healthy subjects.

**Figure 3 biomolecules-09-00355-f003:**
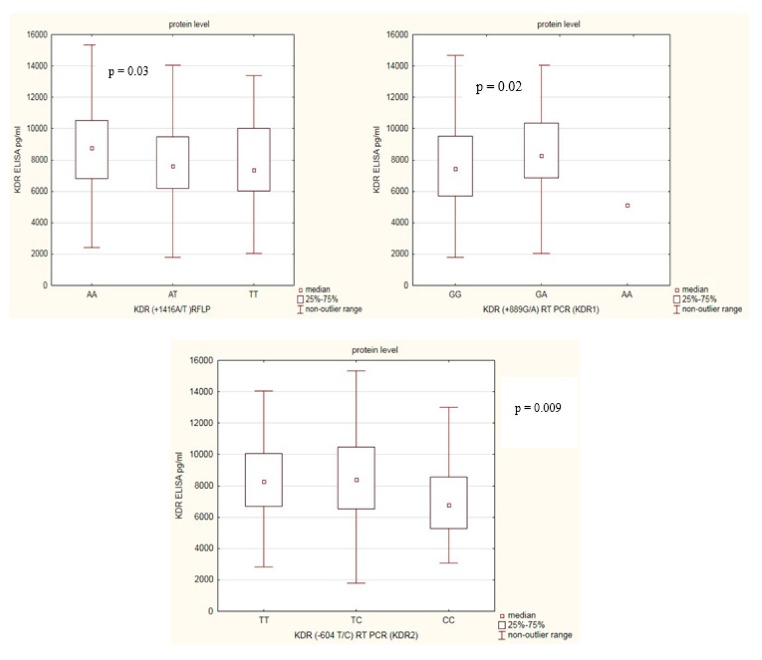
Variation in KDR serum levels in RA patients in relation to *KDR* genotypes.

**Table 1 biomolecules-09-00355-t001:** Demographic and clinical characteristics of patients with Rheumatoid Arthritis (RA).

Characteristics	RA Patients	
	N	Mediana (IQR)
Age (years)	*616*	56.2 ± 12.5 (22–89)
Disease duration (years)	*524*	11.2 ± 8.7 (0–48)
Larsen	*567*	3.0 ± 1.0 (0–5)
Number of swollen joints	*340*	4.6 ± 4.9 (0–26)
Number of tender joints	*338*	8.0 ± 6.1 (0–28)
ESR (mm/h)	*565*	35.0 ± 24.5 (0–164)
CRP (mg/L)	*345*	22.5 ± 24.2 (0–144)
Hemoglobin (g/dL)	*345*	12.5 ± 1.5 (8.1–16.5)
VAS (mm)	*331*	52.3 ± 23.9 (0–100)
DAS 28-CRP	*334*	4.89 ± 1.40 (0.97–7.95)
HAQ	*288*	1.47 ± 0.72 (0–3.125)
PLT (× 10^3^/mm^3^)	*345*	321.17 ± 106.10 (0–948)
Creatinine	*344*	0.73 ± 0.24 (0–2.60)
	**N**	**n (%)**
RF presence	*561*	386 (69%)
anti-CCP presence	*348*	280 (80%)
Coronary artery disease CAD	*343*	46 (13%)
Hypertension (HNT)	*344*	123 (36%)
Myocarditis (MI)	*341*	12 (3.5%)

N—number of patients with clinical information’s; n—number of patients with positive clinical manifestation; DAS-28—disease activity score for 28 joints ,VAS—visual analogue scale (range 0–100), HAQ—Health Assessment Questionnaires (range 0–3), CRP—C-reactive protein, ESR—erythrocyte sedimentation ratio, PLT—platelet, RF—rheumatoid factor, anti-CCP—anti-CCP antibodies. IQR—interquartile range;

**Table 2 biomolecules-09-00355-t002:** Clinical characteristics of patients with cardiovascular diseases.

Parameter *Continuous Variables*	Patients with Cardiovascular Diseases		Patients without Cardiovascular Diseases		*p*
	*N*	Median (IQR)	*N*	Median (IQR)	
age (years)	*145*	62 (56–68)	*199*	53 (44–60)	**0.000**
disease duration (years)	*132*	10 (6–17)	*167*	10 (5–17)	0.932
Larsen	*144*	3 (2–4)	*199*	3 (3–4)	0.913
number of tender joints	*140*	8 (4–13.5)	*193*	7 (2–11)	0.247
number of swollen joints	*141*	3 (0–7)	*194*	3 (1–7)	0.280
ESR (mm/h)	*144*	30 (15.5–45)	*198*	24 (12–40)	0.046
CRP (mg/L)	*141*	15.7 (8.0–36.7)	*198*	12.2 (5.1–32.0)	0.053
Hb	*142*	12.8 (11.7–13.6)	*197*	12.6 (11.5–13.5)	0.431
VAS (mm)	*139*	58 (32–75)	*188*	52 (34–69)	0.539
DAS-28	*139*	5.10 (3.98–6.1)	*191*	5.05 (3.77–5.89)	0.278
HAQ	*123*	1.63 (1.00–2.13)	*158*	1.38 (0.75–1.88)	0.018
PLT	*142*	298.5 (243–362)	*197*	317 (259–386)	0.277
Creatinine	*142*	0.70 (0.60–0.87)	*196*	0.70 (0.60–0.80)	0.006
parameter *discrete variables*	patients with cardiovascular diseases		patients without cardiovascular diseases		*p*
	*N*	n (%)	*N*	n (%)	
women	*145*	133 (92%)	*198*	182 (92%)	0.948
RF +	*144*	102 (71%)	*193*	117 (61%)	0.052
anti-CCP +	*143*	121 (85%)	*197*	153 (78%)	0.110

N—number of patients with clinical information’s; n—number of patients with positive clinical manifestation; *p* < 0.003 was considered significant (according to Bonferroni correction).

**Table 3 biomolecules-09-00355-t003:** Genetic effects of individual Single Nucleotide Polymorphisms (SNPs) in the kinase insert domain-containing receptor (KDR) gene on RA risk.

KDR SNP	Genotype	RA n (%)	Controls n (%)	Adjusted OR (95% CI)	*p*-Value
rs1870377A/T (+1416A/T)					
Codominant	AA	291 (48%)	167 (53%)	1	-
	AT	239 (40%)	120 (38%)	0.78 (0.55–1.09)	0.139
	TT	72 (12%)	26 (8%)	1.76 (1.09–2.85)	**0.020**
Dominant	AA	291 (48%)	167 (53%)	1	
	AT + TT	311 (52%)	146 (47%)	1.12 (0.91–1.39)	0.286
Recessive	AA + AT	530 (88%)	287 (92%)	1	
	TT	72 (12%)	26 (8%)	1.53 (1.07–2.20)	**0.019**
Allele	A	821 (68%)	454 (73%)	1	-
	T	383 (32%)	172 (27%)	1.38 (1.00–1.92)	0.053
rs2305948 G/A (+889G/A)					
Codominant	GG	409 (64%)	245 (72%)	1	-
	GA	224 (35%)	94 (28%)	0.72 (0.21–2.55)	0.614
	AA	8 (1%)	1 (0%)	3.56 (0.30–42.35)	0.314
Dominant	GG	409 (64%)	245 (72%)	1	
	GA + AA	232 (36%)	95 (28%)	1.38 (1.10–1.73)	**0.005**
Recessive	GG + GA	633 (99%)	339 (100%)	1	
	AA	8 (1%)	1 (0%)	2.73 (0.43–17.37)	0.287
Alleles	G	1042 (81%)	584 (86%)	1	-
	A	240 (19%)	96 (14%)	1.71 (1.15–2.54)	**0.008**
rs2071559 T/C (−604 T/C)					
Codominant	TT	163 (26%)	17 (5%)	1	-
	TC	335 (52%)	239 (70%)	0.51 (0.37–0.69)	**0.000**
	CC	141 (22%)	84 (25%)	0.57 (0.39–0.81)	**0.002**
Dominant	TT	163 (26%)	17 (5%)	1	
	TC + CC	476 (74%)	323 (95%)	0.39 (0.27–0.55)	**0.000**
Recessive	TT + TC	498 (78%)	256 (75%)	1	
	CC	141 (22%)	84 (25%)	0.90 (0.71–1.14)	0.360
Alleles	T	661 (52%)	273 (40%)	1	-
	C	617 (48%)	407 (60%)	0.60 (0.45–0.81)	0.001

**Table 4 biomolecules-09-00355-t004:** KDR haplotypes in rheumatoid arthritis patients and controls.

Haplotype 1416A/T 889G/A 604T/C	RA 2n = 1200 (freq)	Control 2n = 622 (freq)	*p*-Value	Odds Ratio (95%CI)
AGC	449 (0.374)	310 (0.498)	**<0.001**	0.601 (0.494–0.731)
TAT	116 (0.096)	47 (0.075)	0.141	1.309 (0.919–1.864)
TGT	154 (0.128)	67 (0.107)	0.225	1.219 (0.899–1.654)
AGT	286 (0.238)	107 (0.172)	**0.001**	1.506 (1.176–1.927)
TGC	85 (0.07)	47 (0.075)	0.704	0.932 (0.644–1.35)
AAT	66 (0.055)	23 (0.036)	0.108	1.515 (0.933–2.461)

**Table 5 biomolecules-09-00355-t005:** Association between genotypes of KDR +889 G/A and clinical characteristics among RA patients.

Parameter	GG		GA + AA		*p* *
	*N*	Median (IQR)	*N*	Median (IQR)	
Age (years)	631	50 (32–59)	23	52 (33–62)	0.737
Disease duration (years)	322	10 (4–16)	202	9 (5–15)	0.588
Larsen	355	3 (3–3)	211	3 (3–4)	0.030
Number of tender joints	175	8 (4–12)	163	6 (2–11)	0.062
Number of swollen joints	176	4 (2–9)	164	2 (0–6)	**0.000**
ESR (mm/h)	355	31 (19–51)	209	28 (15–40)	0.006
CRP (mg/L)	177	17 (8–40)	167	11 (5–25)	**0.000**
Hemoglobin (g/dL)	177	12.6 (11.6–13.3)	167	12.8 (11.7–13.8)	0.201
VAS (mm)	170	60 (45–75)	161	47 (30–67)	**0.000**
DAS 28-CRP	172	5.3 (4.1–6)	>162	4.7 (3.5–5.6)	**0.000**
HAQ	>152	1.62 (1.06–2)	135	1.37 (0.75–2)	0.027
PLT (× 10^3^/mm^3^)	177	312 (255–383)	166	304 (245–370)	0.567
Creatinine	174	0.7 (0.6–0.8)	167	0.7 (0.6–0.8)	0.027
					*p* **
	*N*	n (%)	*N*	n (%)	
Women	*635*	481 (76%)	*322*	277 (86%)	**0.000**
RF presence	*352*	241 (68%)	*208*	144 (69%)	0.850
anti-CCP presence	*179*	141 (77%)	*168*	138 (72%)	0.429

N—number of patients with clinical information’s; n—number of patients with positive clinical manifestation; IQR—interquartile range; *p* *—U Mann-Whitney test; *p* **—χ^2^ test; *p* < 0.003 was considered significant (according to Bonferroni correction).

**Table 6 biomolecules-09-00355-t006:** Correlation of KDR protein concentration of the various clinical parameters (RA).

Parameter		KDR Protein Level			KDR Protein Level		*p*
	*Parameter Group I*	*N*	Median (IQR)	*Parameter Group II*	*N*	Median (IQR)	
age	age ≥ 56	*146*	8042 (6575–9935)	age < 56	*126*	7893 (5836–10246)	0.337
sex	women	*253*	7955 (6226–9848)	men	*19*	9867 (6575–10910)	0.093
RF	RF +	*169*	7964 (6042–10032)	RF -	*97*	8457 (6595–10246)	0.413
anti-CCP	a-CCP +	*207*	8006 (6294–10032)	a-CCP -	*58*	8254 (6438–10323)	0.586
disease duration	≥10	*151*	8065 (6429–10032)	<10	*113*	7825 (6101–10140)	0.894
ESR	≥30	*124*	8269 (6434–9966)	<30	*141*	7825 (6042–10128)	0.420
number of tender joints	≥7	*155*	7415 (5852–9997)	<7	*107*	8370 (6892–10360)	0.081
number of swollen joints	≥3	*141*	7375 (5836–9997)	<3	*121*	8449 (7008–10323)	0.081
CRP	≥13	*121*	8006 (6006–9518)	<13	*143*	8091 (6556–10450)	0.191
DAS-28	≥5.0	*127*	7791 (5894–9997)	<5.0	*133*	8032 (6556–10246)	0.508
HAQ	≥1.5	*126*	7722 (5836–10141)	<1.5	*113*	7887 (6556–9642)	0.789
cardiovascular diseases	CAD + (CAD, HNT, MI)	*117*	7791 (6423–10360)	CAD -	*145*	8032 (6303–9848)	0.889

N—number of patients with clinical information’s

**Table 7 biomolecules-09-00355-t007:** Variation in KDR expression levels in RA patients and the control group in relation to *KDR* gene polymorphisms.

Genotype	RA Group		Control Group		*p*
	*N*	Median (IQR)	*N*	Median (IQR)	
KDR (+1416A/T)					
AA	*127*	8766 (6802–10495)	*146*	7479 (5297–9181)	**0.001**
AT	*96*	7611 (6199–9457)	*97*	7527 (5341–8963)	0.512
TT	*39*	7365 (6042–9997)	*20*	6997 (5322–9204)	0.325
KDR (+889G/A)					
GG	*124*	7425 (5681–9539)	*205*	7384 (5348–8973)	0.398
GA	*147*	8264 (6875–10360)	*83*	7371 (5469–8919)	**0.002**
AA	*1*	-	*1*	-	-
KDR (−604 T/C)					
TT	*68*	8269 (6683–10043)	*17*	8917 (7416–9312)	0.934
TC	*152*	8410 (6511–10455)	*200*	7336 (5341–8826)	**0.001**
CC	*51*	6786 (5282–8568)	*72*	6979 (5139–9288)	0.900

N—number of patients.

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
