# Peer review of "KDR (VEGFR2) Genetic Variants and Serum Levels in Patients with Rheumatoid Arthritis"

_biomolecules, 2019, doi:10.3390/biom9080355_

Round 1
Reviewer 1 Report
I've read with attention the paper by Paradowska-Gorycka et al. that is potentially of interest. The methodology applied is overall correct, the reported results reliable and adequately discussed. I've only two minor comments:
-The abstract shoud report some quantitative data.
- Figure 2 should reports the significant difference. The same is valid for Figure 5.
Author Response
We deeply appreciate the Reviewer’s professional remarks.
Comment 1. The abstract shoud report some quantitative data.
According to the Reviewer’s suggestion we added in the abstract some quantitative data. "We investigated KDR polymorphisms and protein levels in relation to susceptibility to and severity of RA. 641 RA patients and 340 controls (HC) were examined for rs1870377 KDR variant by PCR-RFLP method and for rs2305948 and rs2071559 KDR SNPs by TaqMan SNP genotyping assay. KDR serum levels were determined by ELISA. The rs1870377 KDR variant has shown association with RA under the codominant (p=0.02, OR=1.76, 95% CI=1.09 – 2.85) and recessive models(p=0.019, OR=1.53, 95% CI=1.07 – 2.20). KDR rs2305948 was associated with RA under the dominant model (p=0.005, OR=1.38, 95% CI=1.10 – 1.73). Under the codominant model, the frequency of the rs2071559 TC and GG genotypes were lower in RA patients than in controls (p<0.001, OR=0.51, 95% CI=0.37 – 0.69, and p=0.002, OR=0.57, 95% CI=0.39 – 0.81). KDR rs2071559 T and rs2305948 A alleles were associated with RA (p=0.001, OR=0.60, 95% CI=0.45 – 0.81 and p=0.008, OR=1.71, CI=1.15 – 2.54). KDR rs2305948SNP was associated with DAS-28 score (p<0.001), VAS score (p<0.001), number of swollen joints (p<0.001), mean value of CRP (p<0.001). A higher KDR serum level was found in RA patients than in HC (8018 pg/ml vs 7381 pg/ml, p=0.002). Present results shed light on the role of KDR genetic variants in the severity of RA".
Comment 2. Figure 2 should reports the significant difference. The same is valid for Figure 3.
Many thanks for Reviewer’s suggestions. We corrected both Figures and added information about p-value. Please refer to the revised version of the manuscript.

Reviewer 2 Report
In this article "KDR (VEGFR2) genetic variants and serum levels in patients with Rheumatoid Arthritis". The authors examined the KDR (also call VEGFR2) SNP and serum levels in RA. This work is well designed and provide new in RA progression.
Minor points:
The ligand of VEGFR2 such as VEGFC SNP should be mentioned and discussed. The limintation should be discussed The results of Fig 2 and Fig 3 are not significant. This part should be discussed.
Author Response
Thank you very much for reviewer comments.
Comment 1. The ligand of VEGFR2 such as VEGFC SNP should be mentioned and discussed.
According to the Reviewer’s suggestion in discussion we added some information about VAGF-A and VEGF-C SNPs. "Also, several genetic variants located in the VEGF-A gene have shown association with rheumatoid arthritis, cancer, coronary artery disease, chronic obstructive pulmonary disease. The rs1570360, rs699947, rs2010963, rs833070 and rs3025030 VEGF gene polymorphisms were associated with RA [23, PMID: 26825024 ], cancer [PMID: 30977010, PMID: 30849545, PMID: 30609111, PMID: 30397360, PMID: 28039484], heart disease [PMID: 30689460], chronic obstructive pulmonary disease [PMID: 27163696, PMID: 27019442] or type 2 diabetes [PMID: 29533820]. While the VEGF-C gene polymorphisms were not examined in patients with RA, they were examined in patients with cancer and Kawasaki disease. VEGF-C rs7664413, rs2046463 and rs1485766 SNPs have shown association with cancer [PMID: 24608123, PMID: 23593187, PMID: 24478168 ], -634 G/C have shown association with Kawasaki disease [PMID: 17874221]".
Please refer to the revised version of the manuscript.
Comment 2. The limitation should be discussed
As the Reviewer suggested, the limitation of the present study was discussed.
"First we checked for genotype call errors. The genotyping error minimization was achieved by genotyping repeated on randomly 20% selected samples (10% for RA patients and 10% for healthy subjects), giving complete conformity of the result. Second, our sample size is relatively small and this may lead to genetic drift, which can result in a loss of polymorphism and driving the frequency of one allele to 1. However, in the case of subjects, withdrawal from HWE, assuming that sources of errors have been eliminated, may indicate a genetic association and a connection of the place with the disease".
Please refer to the revised version of the manuscript.
Comment 3. The results of Fig 2 and Fig 3 are not significant. This part should be discussed.
In the Figure 2 and Figure 3 we added the information about p-value. In our study we found some differences: "In RA patients with the KDR rs1870377AA genotype, the KDR serum levels were significantly higher compared with RA patients with KDR rs1870377AT or rs1870377TT genotypes (p=0.03). RA patients with rs2071559CC genotype had the lowest KDR serum levels comparing with RA patients with rs2071559CT or rs2071559TT genotypes (p=0.009). Furthermore, RA patients with rs2305948GA genotype had the highest and RA patients with rs2305948GG genotype had the lowest KDR serum levels (p=0.02)". Please refer to the revised version of the manuscipt.